# Biodegradation of Cholesterol by *Enterococcus faecium* YY01

**DOI:** 10.3390/microorganisms11122979

**Published:** 2023-12-13

**Authors:** Ruimin Yang, Shahbaz Ahmad, Hongyan Liu, Qianqian Xu, Chunhua Yin, Yang Liu, Haiyang Zhang, Hai Yan

**Affiliations:** School of Chemistry and Biological Engineering, University of Science and Technology Beijing, Beijing 100083, China; yrm@st.gxu.edu.cn (R.Y.); shahbazbbt19@yahoo.com (S.A.); lhy55492c@163.com (H.L.); qianqianxu@ustb.edu.cn (Q.X.); chyin@sina.com (C.Y.); liuyang@ustb.edu.cn (Y.L.); zhanghy@ustb.edu.cn (H.Z.)

**Keywords:** cholesterol, *Enterococcus faecium* YY01, biodegradation, genomic analysis

## Abstract

Cholesterol (CHOL) is one of the risk factors causing the blockage of the arterial wall, atherosclerosis, coronary heart disease, and other serious cardiovascular diseases. Here, a promising bacterial strain for biodegrading CHOL was successfully isolated from the gut of healthy individuals and identified as *Enterococcus faecium* YY01 with an analysis of the 16S rDNA sequence. An initial CHOL of 1.0 g/L was reduced to 0.5 g/L in 5 days, and glucose and beef extract were found to be optimal carbon and nitrogen sources for the rapid growth of YY01, respectively. To gain further insight into the mechanisms underlying CHOL biodegradation, the draft genome of YY01 was sequenced using Illumina HiSeq. Choloylglycine hydrolase, acyltransferase, and alkyl sulfatase was encoded by gene0586, gene1890, and gene2442, which play crucial roles in converting 3α, 7α, 12α-trihydroxy-5β-choranic acid to choline-CoA and then choline-CoA to bile acid. Notably, choloylglycine hydrolase was closely related to the biosynthesis of both primary and secondary bile acid. The findings of this study provide valuable insights into the metabolism pathway of CHOL biodegradation by YY01 and offer a potential avenue for the development of bacterioactive drugs against hypercholesterolemia.

## 1. Introduction

Cholesterol (CHOL) is a naturally occurring substance in the human body and an important component of cell membranes crucial to their structure and maintenance [1]. The liver and intestinal mucosa in the human body can synthesize about 80% of the body’s required CHOL from scratch (endogenous) [2], requiring a small amount (20%) of dietary (exogenous) CHOL. When CHOL in the blood increases, it leads to the deposition of CHOL in blood vessels, resulting in the blockage or narrowing of the arteries supplying blood to the heart and the brain, increasing the risk of cardiovascular disease (CVD) [3]. According to the World Health Organization (WHO), CVD accounts for 30% of global deaths and is expected to remain a major cause of death in the next 20 years. By 2030, CVD will affect nearly 23.3 million people worldwide [4].

Currently, the main drugs used to lower blood CHOL levels are statins and ezetimibe, which may also have adverse side effects in some cases [5]. Therefore, probiotics seem to be a promising alternative as, when an adequate amount of live bacteria drugs is ingested, they can bring benefits to the human body [6]. Lactic acid bacteria (LAB) are important microorganisms used for food fermentation worldwide. In addition, LAB, as probiotics, have become a focus of extensive international research due to their beneficial effects on health, including lowering serum CHOL levels [7]. It has been hypothesized that the exopolysaccharides (EPS) of LAB may absorb CHOL and that, subsequently, CHOL binds to the cell surface of the LAB cells or binds to cell membranes, resulting in the inhibition of CHOL assimilation by intestinal cells [8,9]. In some studies, LAB have shown a high efficiency in biodegrading CHOL, such as *Lactobacillus acidophilus* ATCC33200 [10], *Pediococcus pentosaceus* ENM104 [11], and *L. oris* HMI68 [12], which also showed a strong activity of bile salt hydrolase (BSH). The CHOL-lowering mechanism of *L. plantarum* Lp10 [13] was studied by combining transcriptome analysis and real-time quantitative PCR. *L. plantarum* Pw4 [14] reduced the content of low-density lipoprotein cholesterol (LDL-C) in the serum of rats and increased the content of high-density lipoprotein cholesterol (HDL-C). In another study, *L. plantarum* ECGC13110402 [15] was tested for its CHOL-lowering ability in 16 adults with hypercholesterolemia, and the result showed a 28.4% reduction in LDL-C, a 17.6% reduction in non-HDL-C compared to the placebo, and no adverse effects.

Currently, there are also many studies on *Enterococcus faecium*, such as *E. faecium* MC-5 [16], which has been studied for its probiotic properties. The results showed that *E. faecium* MC-5 had a high BSH activity and the ability to remove 70.27% of CHOL. Valledor et al. [17] screened two strains of *E. faecium*, ST20Kc and ST41Kc, and evaluated their characteristics and safety. In another study, Akbar [18] performed genome sequence analysis on *E. faecium* R9 and found that it can produce three bacteriocins, which can be added to food to extend its shelf-life [19]. However, less information is provided on the isolation of functional *E. faecium* from the gut of healthy individuals.

In some studies of *E. faecium*, BSH has been found and acted with probiotics. BSH is an important enzyme enhancing bile binding that belongs to the Cholylglycine Hydrolase (CGH) family of the N-hydrolase superfamily [20,21,22,23,24,25]. BSH is widely distributed in microorganisms in the gastrointestinal tract of mammals and plays an important role in reducing blood CHOL levels. A major beneficial effect of bile salt deconjugation includes the lowering of serum CHOL levels, as the reduced solubility and recycling of deconjugated bile acids forces an increase in the de novo synthesis of bile acids utilizing CHOL [26]. Bile acids are synthesized de novo by CHOL and are coupled to taurine or glycine in the liver via amide bonds at the carboxyl C-24 position. Recent reports have shown that a decrease in serum CHOL levels is associated with the presence of BSH. The BSH activity of probiotics reduces blood CHOL concentration in patients with hypercholesterolemia, which is one of the functional effects of probiotics. This has led to growing concerns about the possible use of BSH enzymes in the treatment of hypercholesterolemia in humans [27,28].

In this study, a CHOL-biodegrading bacterial strain was isolated from the gut of healthy individuals and identified as *E. faecium* YY01 through the 16S rDNA sequence. The optimal growth conditions of YY01 were explored using the response surface method, which provided a basis for the fermentation and application of YY01. The genes and enzymes involved in the biodegradation of CHOL were discovered, and the biodegradation function of YY01 was confirmed at the gene level. The purpose of this study was to screen a CHOL-biodegrading probiotic strain and provide the possibility for probiotics to replace chemical drugs in the treatment of hypercholesterolemia. These findings lay the foundation for further exploration of the CHOL biodegradation mechanism and are of great significance in the development of CHOL-lowering active microbial drugs.

## 2. Materials and Methods

### 2.1. Samples and Media

CHOL with a purity of 99% was purchased from the Macklin Company (Shanghai, China). All the other reagents used for this analysis were of chromatographic grade. The bacterial strain used in this study was isolated using CHOL as the sole carbon and energy source from the gut of healthy individuals bought from Beijing Fumate Biotechnology Co., Ltd. (Beijing, China).

The mineral salt medium (MSM) used for isolating the bacterial strains contains the following (g/L): NH_4_Cl, 0.5; KH_2_PO_4_, 0.5; Na_2_CO_3_, 0.5; MgSO_4_, 0.1; peptone, 0.5; yeast extract, 0.5; Tween 80, 1 mL; and water, 1 L. CHOL was added to the medium at a certain initial concentration as the sole carbon. And, the initial pH of the medium was adjusted to 7.0 with 1.0 mol·L^−1^ NaOH or HCl. The medium used for cultivating *E. faecium* (MRS, Hopebio, Qingdao, China) contained the following (g/L): peptone, 10.0; beef extract, 10.0; yeast extract, 5.0; ammonium dihydrogen citrate, 2.0; glucose, 20.0; Tween 80, 1.0 mL; sodium acetate, 5.0; KH_2_PO_4_, 2.0; MgSO_4_, 0.58; MnSO_4_, 0.25; and water, 1 L. The media were sterilized at 121 °C for 20 min in a high-pressure autoclave (LDZH-118 100 L Shanghai, China).

YY01 was inoculated into the sterilized culture medium and grown in a 100 mL flask containing 20 mL of liquid medium. The culture condition was 37 °C with a shaking rate of 200 r·min^−1^. The YY01 which was incubated for 24 h was cultured in liquid medium with CHOL as the sole carbon and energy source to determine its degradation capacity.

### 2.2. Isolation of CHOL Biodegrading Strain

Human intestinal flora samples were dissolved in sterile saline. The 1% solution was inoculated into 50 mL of MSM-CHOL liquid medium and incubated at 37 °C for 5 days under strict anaerobic conditions. Every 3–5 days, 1.0 mL of the cultures was subcultured into a new medium with the same culture conditions each time. The concentration of CHOL was 5 g·L^−1^. After 4 weeks, the last continuously diluted culture was plated on MRS agar using the plating method and incubated at 37 °C for 2 days. Individual colonies grown on MRS agar plates were selected and inoculated into a modified medium containing CHOL to test for biodegradability, a process repeated several times until pure strains were isolated.

### 2.3. Identification and Draft Genome Sequencing of YY01

A newly isolated strain, YJ01, was identified by its morphological and physiological traits and its biochemical characteristics, then further identified using the internal transcribed spacer identification (ITS). The morphology of YY01 was observed with a microscope (CX41, Olympus, Tokyo, Japan). YY01 was inoculated into an MRS medium and incubated at 37 °C with a shaking speed of 200 rpm for 24 h. The culture solutions of YY01 were prepared, and bacterial precipitates were collected by the centrifugation of 10,000 rpm at 4 °C for 20 min and sent to Sangon Biotech (Shanghai, China) for sequencing the draft genome [29]. The bacterial genome was sequenced using de novo sequencing technology, and the genome sequence was assembled from scratch using bioinformatics. [30] A PCR was performed using a pair of universal primers: 27F (5′-GAGTTTTGATCCTGTCCAG-3′) and 1492R (5′-GTTACCTT-GTTACGACTT-3′) [31]. The 16S rDNA sequence of the strain was compared and analyzed using BLAST through GenBank and the Ribosomal Database Project. The phylogenetic tree was constructed using the neighbor-joining method in MEGA 6.0 based on the 16S rDNA gene sequence.

### 2.4. Analysis of CHOL Using HPLC

The CHOL was measured using HPLC (Shimadzu LC-20AT, Tokyo, Japan). The samples were centrifuged at 12,000 rpm for 10 min, at which point the supernatant was poured. A total of 20 mL of ethanol was added to the centrifuged sample and sonicated for 10 min. The samples were passed through an aqueous microporous membrane (0.22 μm) and used for detection. Then, a 20 µL solution was extracted and analyzed using HPLC (chromatographic column: Agilent ZORBAX SB–Aq (150 mm × 4.6 mm, 5 μm-Micron); UV detection wavelength: 206 nm; mobile phase: methanol: 100%; flow rate: 1 mL/min; column temperature: 35 °C) [32]. The standard curves of the CHOL were determined with HPLC and the correlation coefficient of regression equation R^2^ = 0.9997, with a good correlation. The peak area corresponded to the concentration of CHOL in the standard working solution by means of linear fitting, combined with the standard curve. The content of CHOL in the bacterial solution samples could be calculated, and the bacterial biodegradation rate of CHOL could be obtained.

### 2.5. Optimization of Cultivation Conditions

This study was based on an MRS medium and explored the optimal growth conditions for YY01 by changing its carbon source, nitrogen source, carbon-to-nitrogen ratio, temperature, and initial pH. The carbon sources selected were glucose, sucrose, and glycerol, while the nitrogen sources selected were peptone, beef extract, yeast extract, and ammonium chloride. The carbon-to-nitrogen ratios were set at 5:1, 10:1, 20:1, and 30:1; the temperatures were set at 32 °C, 37 °C, and 42 °C, and the initial pH levels were set at 5.2, 6.1, 6.8, and 7.5. Samples were taken every 4 h in each experiment, and the growth was characterized using OD_600nm_ values.

Based on the analysis of single-factor experiments, response surface experiments were designed using the Design-Expert^®^ program (version 10.0.1.0; Stat-Ease, Inc., Minneapolis, MN, USA) [33]. The carbon-to-nitrogen ratio, cultivation temperature, initial pH, and inoculation amount were selected as the influencing factors for the growth of YY01. The experimental combination was designed using Design-Expert^®^. The OD_600nm_ value was input as the response value into the Design-Expert^®^ experimental combination result table, and the experimental data were analyzed through a Design-Expert^®^ analysis. Response surface plots were drawn based on the experimental results by Design-Expert^®^ to obtain regression equations, optimal solutions, and predicted values. The experimental values were compared with the predicted values based on the optimal solutions.

## 3. Results and Discussion

### 3.1. Isolation and Identification of CHOL Biodegrading Strain

The monoclonal colonies of YY01 were grown on the MRS agar plate (Figure 1a) and were milky white and round, with a neat edge and a diameter of 1–2 mm. The morphology of YY01 was observed under a light microscope with 1000× magnification, appearing spherical (Figure 1b).

The nucleic acid sequence of YY01 was amplified and sequenced to determine its phylogenetic placement. According to Figure 2, the association between YY01 and other closely related members reveals YY01′s closest resemblance to *E*. *faecium*. The strain was identified as *E. faecium* YY01 based on a phylogenetic analysis of the 16S rDNA sequence. *E. faecium* has been recently reported to biodegrade CHOL. For example, *E. faecium* [34] was studied in mice on a high-CHOL and high-fat diet, and the results showed a CHOL removal rate of 46.13%.

### 3.2. Biodegradation of CHOL by YY01

An initial CHOL of 1.0 g/L was biodegraded to 0.5 g/L within 5 days by *E*. *faecium* YY01, with a removal rate of 50 ± 0.28% (Figure 3). In other studies, CHOL had been shown to be well biodegradable by *E. faecium*. Albano et al. [35] confirmed that *E. faecium* VC223 displayed a significantly strong removal of CHOL (45%) within 24 h in vitro. In another study, *E*. *faecium* LR13 [36] was screened from soil, and the researchers investigated its BSH activity, ability to biodegrade CHOL in vitro, and probiotic characteristics. The results showed that this strain could survive at a low pH and had a CHOL removal rate of 75.97 ± 1.22%. Compared to the other reported CHOL-biodegrading strains, YY01 is a promising bacterial strain from the gut of healthy individuals for the efficient biodegradation of CHOL. The difference is that the CHOL used in *E*. *faecium* VC223 and LR13 was water-soluble, while the CHOL used in this study was not. Water-insoluble CHOL was thought to be more difficult to be biodegraded than water-soluble CHOL.

### 3.3. The Influence of Cultivation Conditions on the Growth of YY01

The influence of different carbon sources, nitrogen sources, carbon-to-nitrogen ratio, initial pH, and cultivation temperature on the growth of YY01 was studied (Figure 4). Figure 4a shows that glucose, as the optimal carbon source, has a strong promoting effect on the growth of YY01. Figure 4b shows that beef extract significantly promotes the growth of YY01 compared to the other three nitrogen sources, as the optimal nitrogen source. YY01 hardly grew in inorganic nitrogen sources. Figure 4c shows that the optimal carbon-to-nitrogen ratio is 5:1, and, as the value increases, the promotion of YY01 growth decreases. Figure 4d demonstrates the influence of the initial pH of the culture medium on YY01. The results show that a higher initial pH of the culture medium is more favorable for the growth of YY01. As the initial pH decreases, the growth rate of YY01 slows down, presumably meaning that a low pH may inhibit the growth of YY01. Figure 4e shows that the optimal growth temperature for YY01 is 37 °C.

The response surface experimental design and the corresponding results are shown in Table 1. Through the analysis of the data by Design-Expert^®^, the analysis results are shown in Table 2. A higher F value means a more robust model, and a lower *p*-value means it is more significant. A *p*-value of less than 0.05 indicates an important factor [33]. The F values for C/N, initial pH, and temperature were relatively large, with *p* values < 0.004, indicating that these three factors had a significant impact on the growth of YY01. The inoculation amount was not found to have had a significant impact on YY01. From the results it can be concluded that the order of factors affecting the growth of YY01 is as follows: initial pH > C/N > temperature > inoculation amount. Through a polynomial regression analysis, the experimental result data were regressed and fitted, and the regression fitting equation for YY01 liquid fermentation was established as follows:OD_600nm_ = 0.64 − 0.21(C/N) + 0.23(initial pH) − 0.10(temperature) + 0.13(C/N)^2^ − 0.086(temperature)^2^

The optimal fermentation conditions obtained were tested. The best combination was 5.3 of C/N, 7.9 of the initial pH, 33.5 °C in temperature, and two for the inoculation amount. The results are shown in Table 3, and the actual results were consistent with the predicted values. Through an analysis of the response surface plot (Figure 5), it was found that the smaller the C/N ratio and the higher the initial pH, the better the growth of YY01. As the temperature increased, the growth of YY01 first rose and then declined, while the inoculation amount had no effect on growth.

### 3.4. Genomic Analysis and Metabolism Pathway for CHOL Biodegradation

To delineate the mechanism of CHOL removal by YY01, the draft genome was sequenced using the Illumina Hiseq platform with paired-ends sequencing. It revealed a total length of 9.4 Mbp, with an average GC content of 39.01%. The reads were assembled into 64 scaffolds with an N50 of 0.11 Mbp. A total of 2657 protein-coding genes, 64 tRNA genes, and 6 rRNA genes were predicted.

The annotation results of each database were summarized at the gene level to realize multidimensional data mining of genes. Based on protein sequence alignment, gene sequences were compared with each database to obtain the corresponding functional annotation information. The results of genome annotation revealed that 77.23% of genes (1957) were classified into 20 different categories of COG (Figure 6a). Notably, 261 genes were associated with carbohydrates transport and metabolism (G); 129 genes were associated with amino acid transport and metabolism (E); 160 genes were associated with transcription (K), and 108 genes were associated with inorganic ion transport and metabolism (P).

In addition, 1826 genes of *E. faecium* YY01 were annotated in the GO database (Figure 6b). The horizontal coordinate represented the three branches of GO and a further level-2 classification, and the vertical coordinate represented the relative proportion of genes. There were 4151 genes related to biological processes, accounting for 40.54%. There were 3425 genes related to cellular components, accounting for 33.45%. And, there were 2664 genes related to molecular function, accounting for 26.01%. The biological process included the metabolic process, the cellular process, the single-organism process, the regulation of biological processes, the establishment of localization, and other function-related genes. The cellular components included the extracellular region, the cell part, the membrane, the macromolecular complex, the membrane part, and other function-related genes. The molecular function included nucleic acid-binding transcription factor activity, protein-binding transcription factor activity, catalytic activity, transporter activity, molecular transducer activity, and other function-related genes.

Additionally, 1089 genes of *E. faecium* YY01 were annotated in the KEGG database (Figure 6c). The horizontal coordinate represented the level-2 classification of the KEGG pathway, and the vertical coordinate represented the number of genes annotated under this classification. The colors of the columns represented the level-1 classification of the KEGG pathway. And, among them, 359 genes were associated with carbohydrate metabolism, 217 genes with membrane transport, 131 genes with amino acid metabolism, and 125 genes with nucleotide metabolism. In the metabolic classification, amino acid metabolism, carbohydrate metabolism, metabolism of cofactors and vitamins, energy metabolism, nucleotide metabolism, glycan biosynthesis and metabolism, biosynthesis of other secondary metabolites, xenobiotics biodegradation and metabolism, lipid metabolism, metabolism of terpenoids and polyketides, and other related metabolic pathways were annotated.

In the CAZy annotation (Figure 7), a total of 124 genes encoding carbohydrate-active enzymes were found, including Glycosyl Transferases (23 genes), Carbohydrate Esterases (24 genes), Glycoside Hydrolases (56 genes), Auxiliary Activities (6 genes), Carbohydrate-binding Modules (9 genes), and Polysaccharide Lyases (6 genes). Cholesterol is a derivative of cyclopentane polyhydrophenanthrene. According to the CAZy annotation, there are 24 genes encoding Carbohydrate Esterases, which may hide other enzymes and genes involved in cholesterol metabolism. *E*. *faecium* YY01 could be applied in the prevention and treatment of hypercholesterolemia in some ways based on the above assumptions. In another study, Carbohydrate Esterases may have been related to the metabolism of triglycerides, and YY01 could be applied in probiotic therapy for hypertriglyceridemia.

Genes related to bacteriocins and peroxide products were discovered in COG (Table 4). Bacteriocins have been reported to inhibit various clinical pathogenic bacteria and multidrug-resistant bacteria, thereby preventing infections caused by these bacteria in the human body [37]. The discovery of genes related to bacteriocin production in YY01 better demonstrates the research value of YY01 compared to the reported *E. faeciums* with CHOL biodegradation function.

In the recent study, *E. faecium* YY01 was found to be able to biodegrade CHOL. Genes and enzymes involved in CHOL metabolism were discovered (Table 5). In the KEEG database, genes encoded by genes 0586 and 0600 involved in choloylglycine hydrolase were found. Choloylglycine hydrolase is one of the BSH [38] that is important in the biodegradation of CHOL. In this study, Choline-CoA was converted to bile acid through the action of choloylglycine hydrolase (Figure 7). In the COG database, genes 2441 and 2442 involved in alkyl sulfatase were found. In the gene annotations, alkyl sulfatase interacted selectively with steroids containing a hydroxy group. Acyltransferase was encoded by gene1890 and gene2024, which are involved in the final step of bile acid synthesis [39,40]. The reported biodegradation pathway of CHOL was determined (Figure 8) and CHOL was biodegraded to 3α,7α,12α-trihydroxy-5β-choranic acid by a series of enzymes [41,42,43]. Bile acids were conjugated to an amino acid via the formation of a thioester intermediate, which was catalyzed by the bile acid cholyl-CoA synthetase and bile acid CoA, which are amino acid N-acyltransferase enzymes [44]. Finally, choline-CoA was converted to bile acid by means of choloylglycine hydrolase.

In some studies of *E*. *faecium,* plasmids, cytolysin, gelatinase, and biofilm activities were not detected in both *E*. *faecium* YY01 and *E*. *faecium* RI 51 [45]. In another study, the safety of *E*. *faecium* was verified through acute oral administration in mice, and it was found to have no adverse effects on general health. In addition, it was found that *E*. *faecium* WEFA23 [46] significantly reduced the body weight, blood lipid levels (total CHOL, triglycerides, and low-density lipoprotein CHOL), and blood glucose levels in rats. Compared to the reported strains, YY01 was confirmed to be highly effective in biodegrading CHOL in vitro and could be studied in mouse experiments in order to explore the effects of lowering CHOL in the body.

## 4. Conclusions

*E. faecium* YY01, an efficient bacterial strain for biodegrading CHOL, was successfully isolated from healthy human gut. An initial CHOL of 1.0 g/L was reduced to 0.5 g/L in 5 days at 37 °C by YY01. Glucose and beef extract were found to be optimal carbon and nitrogen sources for the rapid growth of YY01. Importantly, a genomic analysis revealed the presence of gene0586, gene1890, and gene2442 encoding choloylglycine hydrolase, acyltransferase, and alkyl sulfatase, which were involved in the biodegradation of CHOL. Notably, choline-CoA was converted to bile acid through the action of choloylglycine hydrolase, and alkyl sulfatase interacted selectively with steroids containing a hydroxy group. These findings revealed the metabolic pathway for the biodegradation of CHOL by YY01 and opened a new avenue for future research on the regulation of CHOL metabolism. These results will provide valuable insights into the theoretical basis for an adjuvant therapy with bacterioactive drugs against hypercholesterolemia.

## Figures and Tables

**Figure 1 microorganisms-11-02979-f001:**
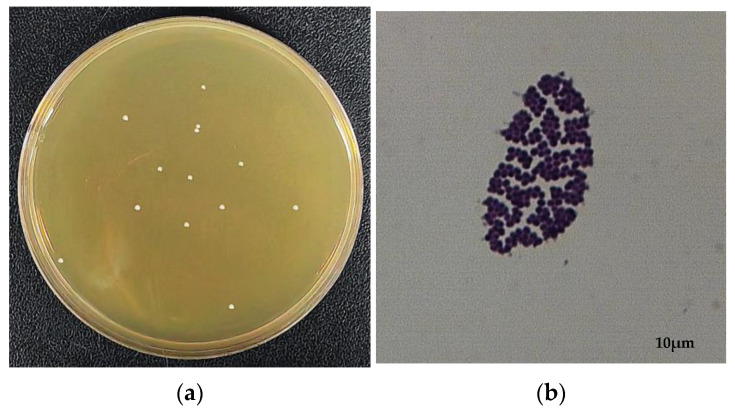
*E. faecium* YY01 colonies (**a**) and morphology (**b**).

**Figure 2 microorganisms-11-02979-f002:**
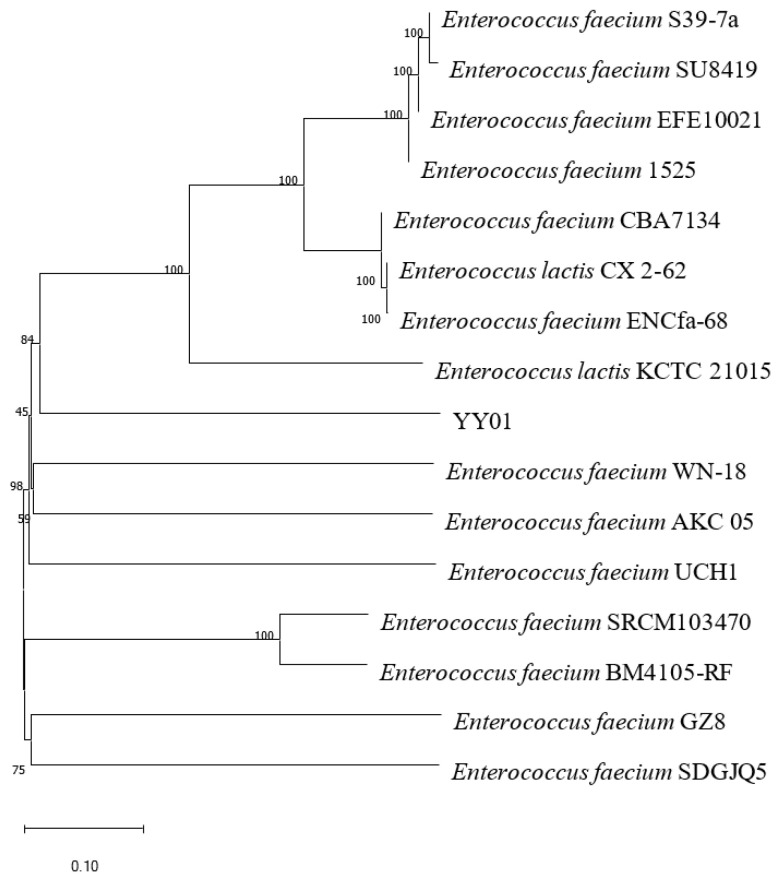
Phylogenetic tree of *E*. *faecium* YY01 based on the 16S rDNA sequence. The phylogenetic relationships between the species were assessed at the genomic level using average nucleotide identity (ANI) analysis. Through this analysis, the identity between strain YY01 and *E*. *faecium* is 84%.

**Figure 3 microorganisms-11-02979-f003:**
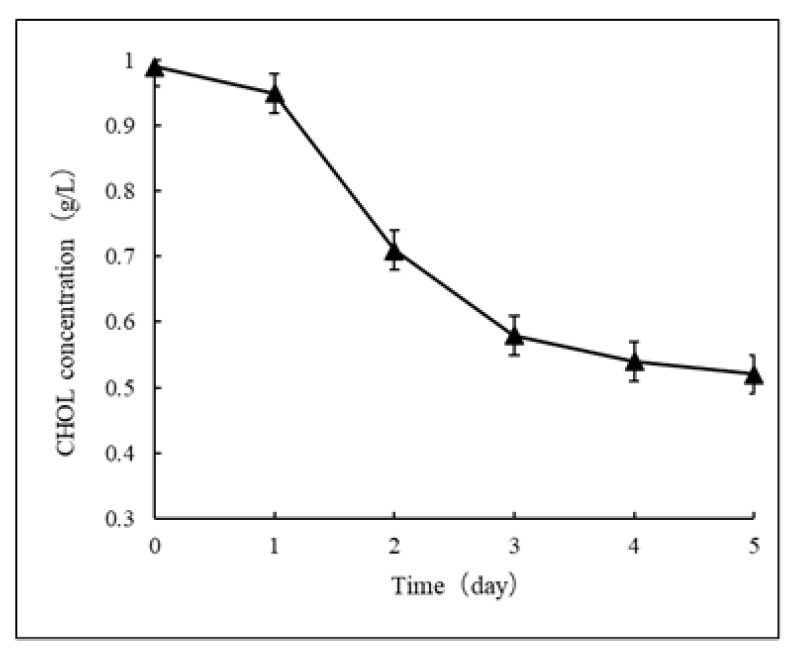
CHOL biodegradation kinetic curve by *E*. *faecium* YY01. CHOL was biodegraded by YY01 from an initial concentration of 1 g/L to 0.5 g/L. The results shown are the mean values from three replicate experiments.

**Figure 4 microorganisms-11-02979-f004:**
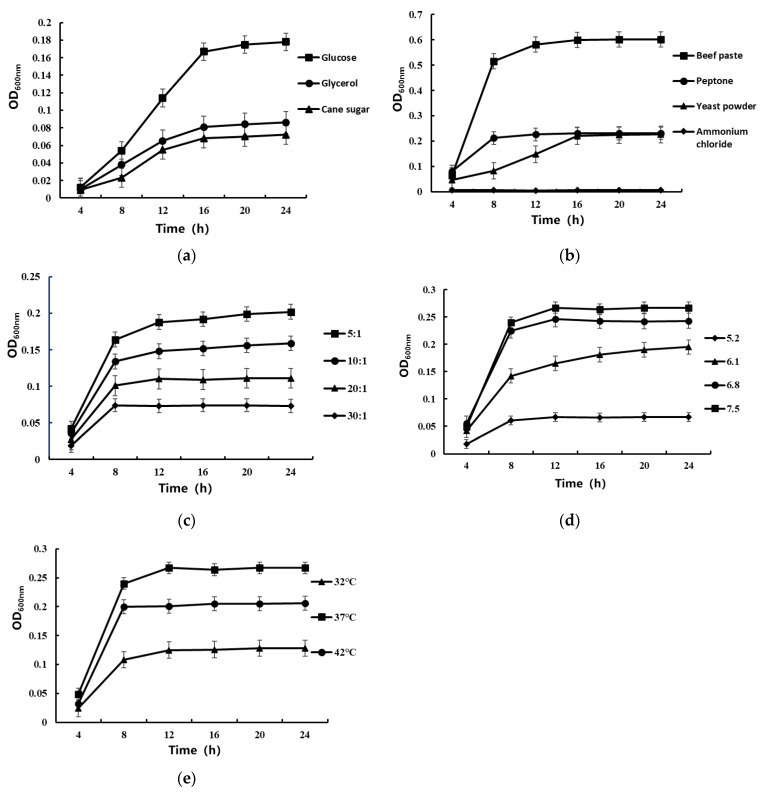
Effects of carbon source (**a**), nitrogen source (**b**), C/N (**c**), initial pH (**d**), and temperature (**e**) on YY01 growth.

**Figure 5 microorganisms-11-02979-f005:**
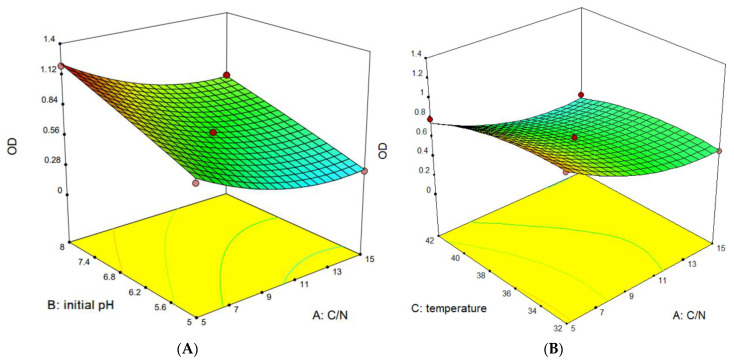
Response surface plot for 24 h of YY01 incubation (C/N (**A**), initial pH (**B**), temperature (**C**), and inoculation amount (**D**)).

**Figure 6 microorganisms-11-02979-f006:**
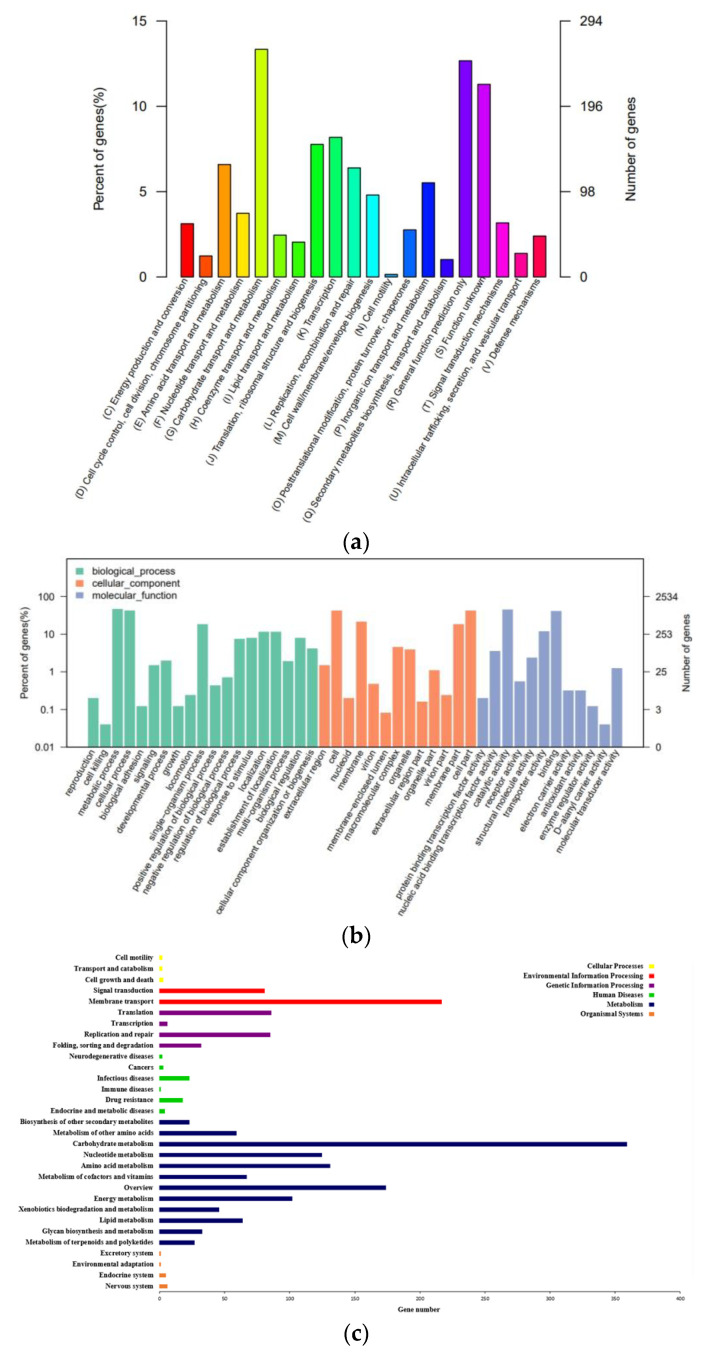
COG (**a**), GO (**b**), KEEG (**c**) annotation classification of *E*. *faecium* YY01.

**Figure 7 microorganisms-11-02979-f007:**
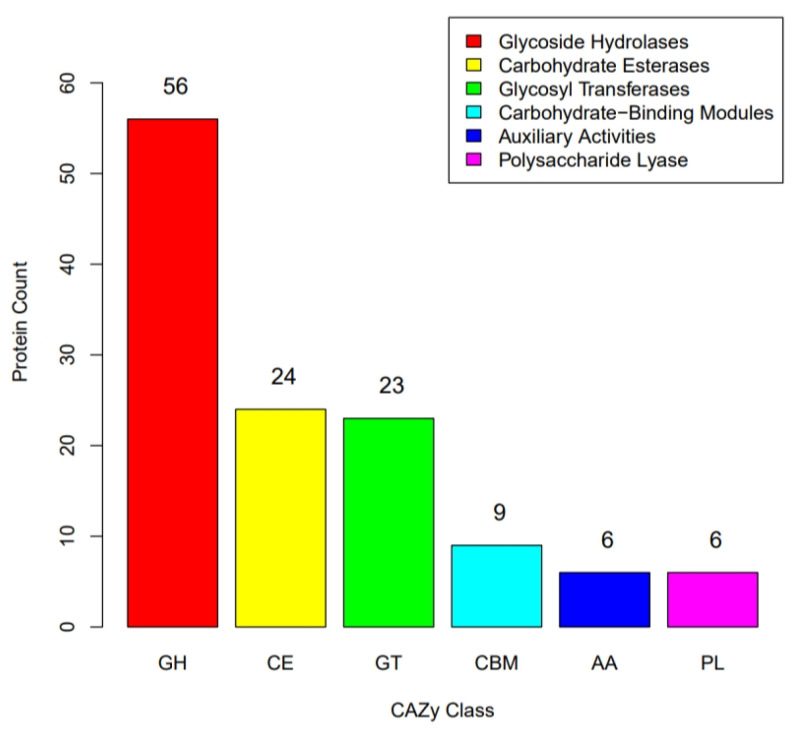
CAZy classification results of *E. faecium* YY01.

**Figure 8 microorganisms-11-02979-f008:**
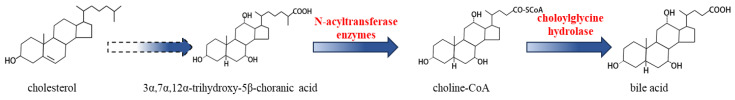
The metabolism pathway for biodegrading CHOL with *E*. *faecium* YY01.

**Table 1 microorganisms-11-02979-t001:** Response surface experimental design and response values of YY01 incubation.

Run	C/N	Initial pH	Temperature	Amount of Inoculation	OD_600nm_
1	10	5.2	42	2	0.070
2	10	6.7	37	2	0.639
3	10	7.5	37	1	0.810
4	10	6.7	37	2	0.639
5	15	7.5	37	2	0.779
6	15	6.7	32	2	0.538
7	10	6.7	42	3	0.498
8	10	7.5	37	3	0.808
9	10	5.2	32	2	0.453
10	10	6.7	37	2	0.639
11	10	6.7	42	1	0.477
12	15	6.7	37	1	0.533
13	5	6.7	37	1	1.008
14	10	7.5	32	2	0.987
15	10	7.5	42	2	0.683
16	15	6.7	37	3	0.594
17	5	6.7	42	2	0.792
18	10	6.7	37	2	0.639
19	10	6.7	32	3	0.552
20	5	6.7	37	3	0.994
21	10	6.7	37	2	0.639
22	15	6.7	42	2	0.445
23	5	6.7	32	2	1.022
24	5	5.2	37	2	0.682
25	10	6.7	32	1	0.636
26	15	5.2	37	2	0.330
27	10	5.2	37	1	0.495
28	10	5.2	37	3	0.475
29	5	7.5	37	2	1.201

**Table 2 microorganisms-11-02979-t002:** Analysis of variance for the cultivation of YY01 during the response surface design.

	Sum of		Mean	F	*p*-Value	
Source	Squares	df	Square	Value	Prob > F	
Model	1.49	14	0.11	21.54	<0.0001	significant
A-C/N	0.51	1	0.51	103.86	<0.0001	
B-pH	0.64	1	0.64	128.92	<0.0001	
C-Temperature	0.12	1	0.12	25.26	0.0002	
D-inoculation amount	0.00012	1	0.00012	0.024	0.8781	
AB	0.0012	1	0.0012	0.25	0.626	
AC	0.0047	1	0.0047	0.95	0.346	
AD	0.0014	1	0.0014	0.28	0.6018	
BC	0.0016	1	0.0016	0.32	0.5828	
BD	0.00008	1	0.00008	0.016	0.8999	
CD	0.0028	1	0.0028	0.56	0.4672	
A^2^	0.12	1	0.12	23.72	0.0002	
B^2^	0.0005	1	0.0005	0.1	0.7546	
C^2^	0.048	1	0.048	9.79	0.0074	
D^2^	0.0001	1	0.0001	0.028	0.8704	
Residual	0.069	14	0.0049			
Lack of Fit	0.069	10	0.0069			
Pure Error	0	4	0			
Cor Total	1.56	28				

**Table 3 microorganisms-11-02979-t003:** Predicted and experimental values of the responses at optimum formulation conditions.

Conditions	C/N	Initial pH	Temperature
**Values**	5.3	7.9	33.5
**Experimental Values**	1.162	1.164	1.261
**Average value**	1.196 ± 0.046		
**Predicted Values**	1.212		

**Table 4 microorganisms-11-02979-t004:** Genes related to bacteriocins and peroxides production in YY01.

Gene ID	Database	Enzyme
gene0657	COG	bacteriocins
gene1725		
gene2190		
gene2034		acyltransferase
gene2240		
gene1055		

**Table 5 microorganisms-11-02979-t005:** Genes and corresponding enzymes related to purines’ biodegradation in YY01.

Gene ID	Database	Enzyme
gene0586, gene0600	KEGG	choloylglycine hydrolase
gene2441, gene2442	COG	alkyl sulfatase and related hydrolases
gene2024,	COG	acyltransferase
gene1890		

## Data Availability

No new data were created or analyzed in this study. Data sharing is not applicable to this article.

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
