# Peer review of "Biodegradation of Cholesterol by Enterococcus faecium YY01"

_microorganisms, 2023, doi:10.3390/microorganisms11122979_

Round 1
Reviewer 1 Report
Comments and Suggestions for Authors
The study by Yan R. and colleagues focused on the isolation of a specific strain from the human intestine, identified as Enterococcus faecium YY01, with the ability to degrade cholesterol. The study is well planned and designed. The results look very interesting and open the door for future probiotic treatment of hypercholesterolemia. However, there are some major concerns that need to be addressed:
- - Introduction:
Line 25: please define CHOL for the first time in the text.
Line 51: “…was investigated the …”, please correct with “… was tested for the CHOL-…”
Line 52-54: please correct the sentence “…the result showed that reductions were noted….”. It should be better “the results showed a 28.4% reduction in LDL-C, …”.
Line 56: please add a full stop to the sentence to make it clearer: “…properties. The results showed that….”
Line 68: please correct “solu-bility”
Line 69: “de novo” should be in italics. Please correct throughout manuscript.
Line 69: please correct “synthe-sis”.
Line 70: replace “cholesterol” with the abbreviation used in the manuscript.
Line 72, 73, 74: please correct the words that appear with a hyphen in the middle.
Line 77: please revise the English in this sentence. You can rewrite the sentence as follows: “In this study, a CHOL-biodegrading bacterial strain was isolated….”
Line 77-86: this paragraph might be more appropriate in the abstract or even the conclusion section of the manuscript. The reviewer believes that the introduction should summarize the main aims of the study and state the hypothesis of the study. Please clearly state the aim/s of the study.
- - Material and Methods
Line 159: Please indicate the specific statistical software used for the RSM analysis. In addition, provide the statistical criteria used to consider significant results and the predicted model used to calculate the optimal response. Please indicate the number of replicates used to validate the results. The RSM design should be carefully explained.
- - Results and discussion
Line 174-175: Figure 1 legend: please rewrite using appropriate expressions: i.e. “Enterococcus faecium YY01 colonies (a) and morphology (b) …..”
Line 191: “biodegradation rate of 50%” is not correct. It would be better to display this value in units related to the rate, i.e. mg·mg-1 h-1 or g·L-1·h-1. Or even not to write “biodegradation rate” instead of “percentage of biodegradation”.
Line 196-198: English correction: “YY01 is a… …reported”. Please rewrite this sentence to make it clearer to the reader.
Line 200: Please correct the English. For example, “Water-insoluble CHOL was thought to be more difficult to biodegrade than water-soluble CHOL”.
Line 213: “op-timal”
Line 251: Is “Design-expert” the statistical software? Please refer to it properly and in the material and methods section.
Line 252 -254: in view of the table results, the two-way interaction for pH and inoculum size is not statistically significant. Why did the authors not reevaluate the model after removing the non-significant terms? It is very likely that the new model would explain a higher percentage of the variation, increasing the R2 value. In addition, there is no significant difference when comparing the linear interaction of the variables. Could the authors explain thist?
Line 259: please replace the letters “A”, “B” and “C”, with the names of the variables to make the regression equation clearer. Given this equation, the reviewer has one point about which he/she is not quite sure about it. In the regression equation “D” is omitted; so has the model been reanalyzed without this variable, which has no significant effect? Please, explain properly.
Line 284: The table header should be written uniformly. Please explain better as “Analysis of variance of cholesterol biodegradation by YY01 during the response surface design”. This is just an example, but explain it properly.
Line 287-288: the reviewer finds Table 3 very confusing. First, the authors should properly specify the optimal conditions estimated. One question that arises in view of the results of table 3 is that if the regression equation does not include the term “inoculum size”, why does the optimal value of it appear in table 3? The authors should explain why they chose this value for the inoculum size. Another question is why are there four different experimental values? Looking at the regression equation, given the estimated value of each variable, the model should calculate a unique response value… Please explain this point in more detail.
Line 303: “And”, please correct the capital letter.
Line 349-353: the author should better explain why YY01 could be used for the prevention of hypercholesterolemia. Please, do not just state “according to the CAZy annotation..”, the authors should explain the relationship between the results obtained by the CAZy classification and this claim.
Lines 358-370: the authors have highlighted three different enzyme activities, encoded by different genes. The reviewer wonders if the authors have considered checking these specific enzymatic activities in the bacterial strain under the conditions defined in the manuscript. Just to check if these activities are enhanced.
- - Conclusions:
Line 390: it is better to replace “intestine” with “gut”.
Comments on the Quality of English LanguageQuality of English was specifically mentioned above.
Author Response
Thanks to the reviewers for providing some suggestions for my article, which have greatly benefited me. By answering and revising the questions raised in the article, I have made significant improvements in both writing and research thinking. Thank you again to the reviewers and editors. Below, I will provide a response to the questions raised by the reviewers.

Reviewer 2 Report
Comments and Suggestions for Authors
In the submitted manuscript, the authors isolated the cholesterol-biodegrading enterococcal strain YY01 identified as Enterococcus faecium. The blood high cholesterol has been recognized be one of causes for heart attack and/or stroke, thus probiotic strain that can biodegrade/absorb cholesterol may bring benefits on human health.
Although the concept of the submitted manuscript is of interest and important, however, the manuscript do not seem to satisfy the criteria for publication as a scientific journal. I have concerns to accept the manuscript with following insufficient points:
There are already many reports regarding lactic acid bacteria that show high efficiency in biodegrading/absorbing cholesterol and result in lowering the cholesterol level, thus the authors should provide more information on superiority/differences of strain YY01 against other strains to indicate originality including comparative genome analyses and in vitro/in vivo assays.
In addition, enterococci are surely regarded as probiotics, however, those species are also recognized to be capable of acquiring and transferring antimicrobial resistance, especially vancomycin, thus if the authors will treat the strain YY01 as probiotic strain, antibiotic resistant profile of the strain should be provided according to the EFSA guidelines.
Author Response
Thanks for raising questions about this study. Your question is extremely important for this study. In the future research, we will continue to explore the probiotic properties and safety of YY01. Thank you again for your valuable feedback on this study.

Round 2
Reviewer 1 Report
Comments and Suggestions for Authors
The study by Yan R. and colleagues has been greatly improved. The clarity and comprehension of the manuscript have improved significantly. There are only a few minor details to consider now:
Line 54-55: the reviewer thinks that the sentence is still unclear. Please, correct as: Currently, there are also many studies on Enterococcus faecium, such as E. faecium MC-5, which has been studied for its probiotic properties.
Lines 80-82: Please correct the English in the sentence as follows: “The purpose of this study was to screen a CHOL-biodegrading probiotic strain and …..”
Lines 152-157: thank the authors for providing information on the software used to design and anlyze the RSM. However, it is necessary to properly cite the software. Please specify the version, brand, and the city. For example: Design Expert (version 7.0.0; State-Ease, Inc., City). You can follow the guidelines in the: https://www.statease.com/software/academic/.
Line 191: Please correct as follows: “Compared to the other reported CHOL-biodegrading strains, YY01 is a promising bacterial strain from the gut of healthy individuals for efficient biodegradation of CHOL.”
Line 223: Please revise along the entire manuscript, but if the authors mention Design-Expert, they should indicate “Design-Expert®”.
Line 264: Table 3: after the changes made to the table, the results are much clearer and do not lead to error. Now it is clear that they are the values of the response variable from the experimental runs and the mean, which was not so clear before. The reviewer appreciates the authors for making it easier for the reader to understand.
Comments on the Quality of English LanguageSome sentences should be corrected, as it is indicated above.
Author Response
Dear reviewer, thank you for your suggestions and questions regarding my work. With your help, the quality of my article has been greatly improved. This includes the writing of English sentences and the use of vocabularies. Additionally, some of the questions you raised about my experimental design have deepened my understanding of this research. Overall, thank you for your attention and support to this research!

Reviewer 2 Report
Comments and Suggestions for Authors
I have understood to the author’s opinion, but the revised manuscript seems to be on borderline, thus I will leave it to the editor to judge.
Author Response
Dear reviewer, thanks for your attention and support in my work. Your suggestion is necessary for the study of YY01. With your suggestion, we will explore the probiotic properties of YY01 in future research, including antibiotic resistance and gene migration. Thank you again for your support of this study.